# Depression in Chinese Patients with Cleft Lip and/or Palate: A Preliminary Study

**DOI:** 10.3390/jcm12041366

**Published:** 2023-02-08

**Authors:** Weiyao Xia, Renjie Yang, Yuan Zong, Yichun Yang, Zhuojun Xie, Ting Chi, Bing Shi, Caixia Gong, Hanyao Huang

**Affiliations:** 1State Key Laboratory of Oral Diseases & National Clinical Research Center for Oral Diseases, Department of Oral Maxillofacial Surgery, West China Hospital of Stomatology, Sichuan University, Chengdu 610041, China; 2State Key Laboratory of Oral Diseases & National Clinical Research Center for Oral Diseases, Eastern Clinic, West China Hospital of Stomatology, Sichuan University, Chengdu 610051, China

**Keywords:** PHQ-9, cleft palate and/or lip, psychology, depression, Chinese patients

## Abstract

(1) Objectives: To investigate the difference in prevalence of depression between patients with CL/P (cleft lip and/or palate) and analyze the possible demographic factors that affect the prevalence of depression in Chinese patients with CL/P. (2) Methods: Patients with CL (cleft lip only), CP (cleft palate), and CLP (cleft lip and palate) were included in the study group. Non-CL/P individuals were included in the control group. The Patient Health Questionnaire (PHQ-9) was used to screen the depression of Chinese patients with CL/P. The different proportions of different depression groups between the CL/P group and the control groups were tested by the Fisher–Freeman–Halton test and Bonferroni correction. The scores between the study groups and the control group were analyzed by one-way ANOVA. In the study groups, demographic and clinical data of the patients, including diagnosis (CL, CP, CLP), gender, age, the only child or not, and region were collected to analyze whether they were the possible factors affecting depression through one-way independent-samples *t*-test. Pearson correlation analysis was used to analyze the correlation between monthly family income and depression. (3) Results: 111 and 80 valid questionnaires were collected from the study and control groups, respectively. The mean PHQ-9 score of the study group (5.459 ± 6.082) was relatively higher than the control group (4.362 ± 3.384), and the difference in proportions of depression groups was statistically significant between the CL/P group and the control group (*p* = 0.01), especially in the mild depression (*p* < 0.05) and moderately severe depression groups (*p* < 0.05). Statistically significant differences in PHQ-9 scores were observed between the individuals of different genders (*p* = 0.036) and ages (*p* = 0.007) in patients with CL/P, the individuals who were the only child or not in patients with CL (*p* = 0.007), and the individuals of different ages in patients with CP (*p* = 0.016). (4) Conclusions: The prevalence of depression in Chinese patients with CL/P was different compared with those without CL/P, while gender, age, the only child or not, and region played significant roles in affecting depression psychology.

## 1. Introduction

Patients with cleft lip and/or palate (CL/P) may suffer setbacks in social life due to facial deformity and speaking problems, which may affect their mental states and quality of life [1]. Studies have shown that patients with CL/P had an increased risk of developing major depressive disorder, anxiety disorders, autism spectrum disorders, and other severe learning disabilities [2,3], which could correlate to poor parent–child emotional communication [4], disharmonious attachment between the two [5], and delay in language expression [6]. However, in some other studies [7], patients did not show unusual levels of psychological distress compared with people without craniofacial anormalities. It reminds us to do further research.

Depression is a common psychological illness affecting mood, thought, and physical health [8]. Depression is characterized by low mood, lack of energy, sadness, insomnia, and the inability to enjoy life [9]. It is generally believed that patients with CL/P are at risk of developing depression [3]. Previous studies have identified depression risk factors, including cognition, stress, being female, parental depression, and other sociodemographic factors [10]. Studies have shown that depression was more prevalent in patients with CL/P than in controls, and dissatisfaction with appearance has been a predictor of depression [11,12]. Sleeping disturbance and person–environment fit, often exhibited in patients with CL/P, are significantly associated with depression [13,14]. However, no quantitative analysis of depression in Chinese patients with CL/P was found, and possible influencing factors remain uncertain.

Psychological intervention is crucial in the teamwork treatment of CL/P, for helping the patients coordinate the treatment and integrate into society better. Hence, in the present study, we aimed to seek to investigate the characteristics of depression in Chinese patients with CL/P by applying the Chinese version of the Patient Health Questionnaire (PHQ-9) and analyzing the possible influencing factors which should be used as screening and monitoring variables. We could provide a theoretical basis, assist the psychological intervention targeting these risk factors in the teamwork treatment for CL/P, and improve the therapeutic effects.

## 2. Materials and Methods

### 2.1. Study Design and Settings

We used the PHQ-9 to measure the depression of patients with CL/P and non-CL/P individuals, then compared the scores to discover the prevalence difference between the groups and the depressive level differences between the patients with certain demographic characteristics. The institutional review board of West China Hospital of Stomatology has approved this study (No. WCHSIRB-D-2016-084R1). The article was performed in concordance with the STROBE checklist for cross-sectional studies (Appendix A, STROBE Statement). Patients visiting West China Hospital of stomatology, Sichuan University, from July 2019 to October 2020, were enrolled. To the patients with CL/P, demographic and clinical data were collected, including diagnosis (CL, cleft lip only; CP, cleft palate only; CLP, cleft lip and palate), gender, age, only child or not, region (urban or rural), and monthly family income (CNY less than 500; 500–1000; 3000–5000; 5000–10,000; 10,000–50,000; more than 50,000) [15].

### 2.2. Participants

The inclusion criteria of the study group were as follows:(1)Patients with complete or incomplete unilateral or bilateral non-syndromic cleft lip and (or) palate;(2)Patients who underwent cleft lip repair and (or) palatoplasty;(3)Patients aged 10 years or above.

The inclusion criteria of the control group were as follows:(1)Individuals with no significant facial defects or other major diseases that may seriously affect physical or mental health;(2)Individuals aged 10 years or above.

### 2.3. Measurement of Depression

The PHQ-9 is a nine-question instrument that is simple and highly operable. It is widely used for screening and evaluating depressive symptoms [12] and is suitable for patients of different cultural backgrounds [16,17,18]. The Chinese version of the PHQ-9 is a valid tool for screening and evaluating depression, with good reliability and validity [19].

The items of the PHQ-9 are as follows:

Q1: Little interest or pleasure in doing things?

Q2: Feeling down, depressed, or hopeless?

Q3: Trouble falling or staying asleep or sleeping too much?

Q4: Feeling tired or having little energy?

Q5: Poor appetite or overeating?

Q6: Feeling bad about yourself—or that you are a failure or have let yourself or your family down?

Q7: Trouble concentrating on things, such as reading the newspaper or watching television?

Q8: Moving or speaking so slowly that other people could have noticed? Or so fidgety or restless that you have been moving a lot more than usual? Thoughts that you would be better off dead, or that you would hurt yourself in some way?

Q9: Thoughts that you would be better off dead or thoughts of hurting yourself in some way?

The score for each item is as follows: 0 = none at all, 1 = a few days, 2 = more than a week, 3 = almost every day, and the summed score of the PHQ-9 ranges from 0 to 27. The cut-off points of 5, 10, 15, and 20 represent mild, moderate, moderately severe, and severe depression, respectively [20]. Items 1, 2, 6, 7, and 9 are classified as cognitive symptoms while items 3, 4, 5, and 8 are classified as somatic symptoms [21].

In this study, the questionnaires were distributed to patients by trained volunteers. Every subject finished the questionnaire independently, and the average response time was about 5 min. Completed questionnaires were collected and checked to ensure integrity, objectivity, and accuracy.

### 2.4. Statistical Analysis

Excluding those invalid questionnaires, the valid data were analyzed by SPSS 26.0 in this study, and the count data was represented by mean ± standard deviation (SD). The different proportions of different depression groups between the study groups and the control groups were tested by the Fisher–Freeman–Halton test and Bonferroni correction. The total scores, two sub-domain scores (cognitive or somatic symptoms) and each item score between the study (CL/P, CL, CP, CLP) and control groups were tested by one-way ANOVA, and *p* < 0.05 was considered statistically significant. The results of the PHQ-9 questionnaire in patients (CL/P, CL, CP, and CLP) were taken as the dependent variable, while gender, age, the only child or not, and region were taken as the independent variables for the independent-samples *t*-test, in which *p*-values < 0.05 was considered statistically significant, and *p*-values < 0.01 were highly statistically significant. The association between monthly family income and depression was tested by Pearson correlation analysis.

## 3. Results

### 3.1. Patient Characteristics

A total of 126 PHQ-9 questionnaires were collected from the study group (111 valid, 15 invalid) (Mean age: 17.46 ± 6.01 years), while a total of 81 questionnaires were collected from the control group (80 valid, 1 invalid) (Mean age:18.48 ± 5.16 years). In the study group, most of the participants were male (61.3%), more than half (72.1%) were not the only child, 59.5% were from rural areas, 57.7% were underage, and only 0.9% reached a monthly family income over CNY 50,000 (Table 1).

### 3.2. Depression Status in Each Group

The difference in proportions of depression groups was statistically significant between the CL/P group and the control group (*p* = 0.01), especially in the mild depression (*p* < 0.05) and moderately severe depression groups (*p* < 0.05) (Table 2). However, the mean total PHQ-9 scores or respective item scores in the study and control groups were not statistically different (Table 3).

### 3.3. Factors Influencing the Scores of Patients with CL/P

In the study group (CL/P), the differences in the total scores of the PHQ-9 were statistically significant between individuals of different genders and ages. In patients with CL, a statistically significant difference was demonstrated between the individuals who were the only child and those who were not. In patients with CP, there was also a statistically significant difference in the individuals of different ages. In patients with CLP, no factor was found to cause any difference (Table 4). The results about factors influencing the cognitive/somatic scores and each item score in the CL/P, CL, CP, and CLP groups are shown in Appendix A.

### 3.4. Correlation between Scores and Monthly Family Income

The significant correlation between the scores and monthly family income was only found in the score of item 5 of the CP group, while there were none in the others (Table 5).

## 4. Discussion

Depression is a high prevalence disorder in adolescence, related to later social and health outcomes [22]. Patients with CL/P experience trouble eating, speech dysfunction, and cosmetic defects from birth, so their mental health could be affected. We aimed to gain a deeper insight into the prevalence of depression in Chinese CL/P patients and identify potential influencing factors. In the present study, we initially found that the prevalence of depression in Chinese patients with CL/P was different compared with those without CL/P and gender, age, and the only child or not, could affect depression in the patients.

We analyzed the depression of Chinese patients with CL/P and identified the latent influencing factors by applying the PHQ-9. Our results showed that the prevalence of depression in Chinese patients with CL/P was statistically different compared with those without CL/P. Interestingly, the differences in the proportions of mild depression (20.7% vs. 36.3%) and moderately severe depression (7.2% vs. 0%) were statistically significant. Previous studies have shown that patients with CL/P suffered from major depressive disorders [2,3]. The mental status and quality of life of patients with CL/P could be seriously affected by facial deformity and speaking problems [1], and they might get teased and isolated. However, there were no statistical differences in the total and each item scores of the PHQ-9 between the CL/P group and the control group, which was similar to the result of previous studies that patients with cleft lip and palate and the control group had no significant differences in holistic psychosocial functioning [23,24].

In our study, the PHQ-9 scores of females were generally higher than males in patients with CL/P. There are statistical differences in some items, consistent with the previous studies where females’ prevalence rates were higher than males’ [25,26]. It could be related to those parents of patients with cleft lip and palate usually having a passive attitude towards the illness. The passive attitude and negative emotions might cause the loss of gender recognition in patients. The importance of appearance was higher in females than males and females had a lower evaluation of their appearance than males [27,28], while males could cover their facial defect with a beard [29]. Furthermore, females were more at risk of suffering from emotional distress than males attributed to higher awareness of sentiment and sensitivity [30]. All the reasons above might explain why those female patients showed higher scores than males in depression. Interestingly, in patients with CL/P, our study showed a significant difference between females and males in item 9 (suicide tendency) scores, which was opposite to a previous report that males had more vigorous attempts to kill themselves than females [31]. We speculated that each factor weighted differently to the depression of patients: for patients with cleft lip and palate, their satisfaction with appearance weighted more, and it needed to be identified later.

There were statistically significant differences between adult and underage patients in patients with CL/P in total scores and item 1 (little interest), 3 (sleeping problems), 4 (lack of energy), and 7 (trouble concentration). It was consistent with the previous finding that the depression would gradually worsen with aging [32]. Adult patients with cleft lip and palate might have a deeper understanding of the disease and pay more attention to their appearance and other abnormal morphology and functions brought by cleft lip and palate. At the same time, adult patients might bear more social pressure, and they were more sensitive to the criticism of others. Though in patients with CL, the scores were generally higher in adult patients but the difference was not statistically significant, probably due to the small samples in the CL group.

The scores of the non-only-child were generally higher than those of the only child, and there were statistical differences in the total scores of the PHQ-9 and items 1 (little interest), 2 (feeling down), and 7 (trouble concentration) between the two groups of patients with CL/P. We supposed that the result was related to how the only child and non-only-child grew up in different environments and cultivating modes. Since the raising pressure was lighter, parents of the only child could provide better living conditions for the growth of their kids, so to a certain degree, the one-child families might have a protective effect on cleft lip and palate patients, leading to relatively light depression symptoms.

We also found that urban and rural patients’ depressive tendencies had apparent differences. CP patients showed unique characteristics in their scores compared to others, which might be related to the fewer samples in the CP group. It was demonstrated that congenital malformations such as cleft lip and palate had a higher prevalence in the areas with a lower economy [33]. In these undeveloped areas, traditional and outdated opinions can affect patients and their family members, such as not knowing the illness enough, low social status, low living standards, inactively searching for psychological aid, or lack of psychological assistance service. Thus, all of which were probably closely related to the results in CLP and CP patients, where those from rural regions showed higher scores in depression than those from urban regions.

Family income was one of the influencing factors for patients’ quality of life with CL/P [34] and severity of depression was negatively associated with the household income [35]. Patients would bring their families enormous direct or indirect medical and financial burdens, which might lead to pressure on themselves and caregivers and depression appearing [36]. However, in our study, we did not find the clear correlation between depression and monthly family income in patients with CL/P, which might result from the low frequency in some groups. Interestingly, the monthly family income of the patients with CP was correlated with the item 5 (abnormal food intake) score. It was known that depressive symptoms were common in eating disorders. Research [37,38] has suggested that eating disorders primarily afflicted young adults within high-income countries. Additionally, CP can lead to dysfunction of swallowing of the patients due to the abnormal anatomical structure of the palate and pharynx, which may affect patients’ appetite and normal food intake [39]. It may explain why the severity of the abnormal food intake was associated with the monthly family income in patients with CP.

Based on the results of this study, we could initially establish a view of the mental health status of patients with CL/P in China. Furthermore, clinical psychologists could carry out individually targeted interventions to help patients with their mental health, which could be vital in the teamwork treatment of CL/P and improve the therapeutic quality.

However, this study still had some limitations. The samples of the patients with CP should be enhanced in the future, and the Patient Health Questionnaire (PHQ-9) can only be used as a screening scale for evaluating the psychological depression of patients with cleft lip and palate, rather than a diagnostic tool for depressive symptoms. As this is a preliminary study, the final study should overcome the aforementioned limitations.

## 5. Conclusions

Though the limitations of the present study existed, by preliminarily evaluating the depression of the patients with CL/P, we still provided some evidence that the prevalence of depression in Chinese patients with CL/P was different compared with those without CL/P. Factors such as gender, age and the only child or not can be the predictive factors of depression in patients with CL/P.

## Figures and Tables

**Table 1 jcm-12-01366-t001:** Demographic characteristics of patients with CL/P.

Variable	n	%
*** Diagnosis**		
CL	35	31.5
CP	16	14.4
CLP	60	54.1
**Gender**		
Male	68	61.3
I Female	43	38.7
**Age**		
Underage	64	57.7
Adult	47	42.3
**Only child or not**		
Yes	31	27.9
No	80	72.1
**Region**		
Urban	45	40.5
Rural	66	59.5
**Monthly family income (CNY)**		
Less than 500	12	10.8
500–1000	12	10.8
1000–3000	31	27.9
3000–5000	29	26.1
5000–10,000	15	13.5
10,000–50,000	11	9.9
More than 50,000	1	0.9

* CL, cleft lip only; CP, cleft palate only; CLP, cleft lip and palate.

**Table 2 jcm-12-01366-t002:** The depression of each group.

	No Depression N (%)	Mild Depression N (%)	Moderate Depression N (%)	Moderately Severe Depression N (%)	Severe Depression N (%)	*p*
CL/P	67 (60.4)	23 (20.7) *	11 (9.9)	8 (7.2) *	2 (1.8)	0.010 *
Control group	46 (57.5)	29 (36.3) *	5 (6.3)	0 (0.0) *	0 (0.0)	

* Significant correlation at 0.05 (two-tailed) (*p* < 0.05).

**Table 3 jcm-12-01366-t003:** The scoring differences between each group.

	CL/P	CL	CP	CLP	Control Group	F	*p*
Total	5.459 ± 6.082	5.600 ± 8.146	6.500 ± 4.082	5.100 ± 5.105	4.362 ± 3.384	0.795	0.529
Cognitive	2.766 ± 3.422	2.829 ± 4.592	3.500 ± 2.098	2.533 ± 2.890	2.163 ± 2.161	0.846	0.497
Somatic	2.694 ± 2.920	2.771 ± 3.781	3.000 ± 2.338	0.517 ± 0.725	2.567 ± 2.493	0.601	0.662
1	0.694 ± 0.760	0.743 ± 0.919	0.688 ± 0.602	0.667 ± 0.705	0.713 ± 0.697	0.067	0.992
2	0.559 ± 0.817	0.600 ± 1.006	0.625 ± 0.719	0.517 ± 0.725	0.488 ± 0.064	0.229	0.992
3	0.838 ± 1.014	0.886 ± 1.105	1.062 ± 1.124	0.750 ± 0.932	0.575 ± 0.776	1.476	0.209
4	0.712 ± 0.908	0.771 ± 1.031	0.750 ± 0.856	0.667 ± 0.857	0.850 ± 0.658	0.481	0.750
5	0.686 ± 0.855	0.686 ± 1.105	0.625 ± 0.885	0.683 ± 0.676	0.525 ± 0.711	0.512	0.727
6	0.649 ± 1.211	0.486 ± 1.040	0.938 ± 0.998	0.667 ± 1.349	0.313 ± 0.587	1.908	0.109
7	0.577 ± 0.869	0.600 ± 1.035	0.813 ± 0.834	0.500 ± 0.770	0.513 ± 0.711	0.545	0.703
8	0.468 ± 0.872	0.429 ± 0.979	0.563 ± 0.892	0.467 ± 0.812	0.250 ± 0.563	1.161	0.328
9	0.288 ± 0.679	0.400 ± 0.946	0.437 ± 0.727	0.183 ± 0.431	0.138 ± 0.381	1.857	0.118

**Table 4 jcm-12-01366-t004:** Factors influencing the total PHQ-9 scores of patients with CL/P.

	Gender			Age			Only Child or Not			Region		
	Male	Female	*p*	Underage	Adult	*p*	Yes	No	*p*	Urban	Rural	*p*
CL/P	4.368 ± 4.309	7.186 ± 7.893	0.036 ^#^	4.141 ± 4.902	7.255 ± 7.060	0.007 ^#^	3.968 ± 4.564	6.038 ± 6.511	0.108	4.578 ± 5.154	6.060 ± 6.612	0.209
CL	3.833 ± 4.274	7.471 ± 10.695	0.205	3.095 ± 4.392	9.357 ± 10.888	0.058	1.600 ± 1.838	7.200 ± 9.133	0.007 ^#^	3.133 ± 4.224	7.450 ± 9.859	0.091
CP	4.750 ± 1.708	7.083 ± 4.522	0.158	3.800 ± 1.095	7.727 ± 4.384	0.016 ^#^	6.400 ± 3.975	6.545 ± 4.321	0.950	7.250 ± 4.743	5.750 ± 3.454	0.482
CLP	4.543 ± 4.515	6.929 ± 6.557	0.220	4.763 ± 5.415	5.682 ± 4.581	0.506	4.688 ± 5.400	5.250 ± 5.049	0.709	4.591 ± 5.662	5.395 ± 4.807	0.561

^#^ Significant correlation at 0.05 (two-tailed) (*p* < 0.05).

**Table 5 jcm-12-01366-t005:** Pearson correlation between scores and monthly family income.

	CL/P	CL	CP	CLP
Total	−0.010	0.042	0.466	0.102
1	−0.075	0.043	0.361	−0.025
2	0.035	0.000	0.307	0.167
3	−0.034	0.033	0.479	0.046
4	0.036	0.154	−0.022	0.153
5	0.096	0.110	0.624 *	0.105
6	−0.043	−0.109	0.106	0.018
7	−0.056	0.041	−0.172	−0.014
8	−0.003	0.179	−0.218	0.136
9	−0.015	0.000	0.233	0.115

* Significant correlation at 0.05 (two-tailed) (*p* < 0.05).

## Data Availability

The authors declare that all data supporting the findings of this study are available upon request to the corresponding author.

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
