# Peer review of "Depression in Chinese Patients with Cleft Lip and/or Palate: A Preliminary Study"

_jcm, 2023, doi:10.3390/jcm12041366_

Round 1
Reviewer 1 Report
GENERAL: Some points were not clear to me and that I consider important.
- As it is preliminary study, I think it is important to make clear the purpose of the final study and this excerpt (so that you can assess whether this preliminary publication makes sense).
- I recommend using the STROBE checklist for article writing.
- The study design was not reported by the authors.
- Review the purpose of the study. In the current format, the role of control in the purpose of the study is not clear. Was assessing whether patients with CL/P are more depressive than those without CL/P an objective? I feel that he is not on target because he did not give statistically significant differences, but this is also a result.
ABSTRACT
- Organize writing, starting with eligibility criteria and following the instruments used.
- Line 17: what diagnosis are you referring to?
- it was not reported in the study which statistical analyzes were used.
- The p-value of statistically significant differences must also appear in the abstract.
-Line 25: check the use of the term incidence. It seems to me that the term prevalence is the correct one.
INTRODUCTION:
The introduction is directing the reader towards another objective than the one described, which would be to verify if patients with CL/P would have more depression than patients without CL/P. However, this objective is not described in the text.
- Are there studies that have analyzed which factors are associated with depression in patients with CL/P? Talk more about what already exists in the literature.
Some results are not responding to the objective, so I am in doubt about whether some details of the method (not described) are relevant. I think it is important for the authors to define the role of control in this study and align with the objectives. Therefore, my comments will consider that this study intends to compare the prevalence of depression between patients with and without CL/P.
- Use the STROBE checklist to report your method, I miss some important details
- Describe better about the objectives of the final study, considering that this publication is a preliminary one.
- Study design was not reported at any time.
- Was the place of recruitment of the control group the same as the cases?
- In this study, is there a need to pair the groups (despite the control group being smaller)? Explain more about it in the text.
- What would be the diagnosis described in line 64.
- Describe the response rate.
- Make clear the groups used for analysis (It was not just with or without CL/P) and detail according to the statistical method. Some analyzes were only clear in the results section.
RESULTS
- I suggest grouping the categories of the monthly family income variable, as some of them have a very low frequency.
- Was the analysis of patients with or without cleft performed using depression in a dichotomous way? With and without depression?
- In table 4, the CL group had a borderline P value for age. Discuss more about it.
- I was confused by the scores in table 5. I went back to the method to understand but it does not describe.
DISCUSSION
- Line 161-169. Discuss the findings. What do they mean?
- Line 170. As there are moments when the control group is compared with the CL/P and in others the analysis involves only the CL/P, make it clear that you are reporting the results of individuals with CL/P. (I reinforce the importance of aligning objectives with findings).
- Did the authors not find other references to support the paragraph of my 184?
- Using the term items and not what they mean makes it very difficult to read the discussion. (line 193)
- I am in doubt about the relevance of the result of the analyzes that were justified in the discussion simply as a lack of sample. Should they be there then? Or were the authors unable to discuss their findings? (Would grouping categories not help?)
Author Response
Q1:As it is preliminary study, I think it is important to make clear the purpose of the final study and this excerpt (so that you can assess whether this preliminary publication makes sense).
A1:Thank you for your advice. We have added more clinical implication and emphasized the importance of the study in the introduction. We also indicated the purpose of the final study at the end of the discussion.
Q2: I recommend using the STROBE checklist for article writing.
A2:Thank you for your advise. We have used the STROBE checklist for article writing.
Q3:The study design was not reported by the authors.
A3:Thank you for your suggestions. We have added the study design.
Q4:Review the purpose of the study. In the current format, the role of control in the purpose of the study is not clear. Was assessing whether patients with CL/P are more depressive than those without CL/P an objective? I feel that he is not on target because he did not give statistically significant differences, but this is also a result.
A4:Thanks for your question. In statistical analysis, we demonstrated the role of control which was to assess the prevalence of depression, the prevalence difference and whether the average level of depression in the groups was different. The results were in TABLE 2 and 3.
ABSTRACT
Q5: Organize writing, starting with eligibility criteria and following the instruments used.
A5:Thank you for your advice. We have organized the writing in the abstract.
Q6:Line 17: what diagnosis are you referring to?
A6:Thanks for your question. The diagnosis referred to CL (cleft lip only), CP (cleft palate only) and CLP (cleft lip and palate). The diagnosis was provided by the surgeon majoring in oral diseases and oral maxillofacial surgery in our hospital. We have added the details in the abstract for better understanding.
Q7: it was not reported in the study which statistical analyzes were used.
A7:Thank you for your comment. We have added the statistical analysis methods Tin the abstract.
Q8:The p-value of statistically significant differences must also appear in the abstract.
A9:Thank you for your advice. We have added the p-value in the abstract.
Q9:Line 25: check the use of the term incidence. It seems to me that the term prevalence is the correct one.
A9:Thank you for your advice. Prevalence is the overall number of people who have experienced something like a disease at any point in time. Incidence is the number of new people who have experienced the same thing over a specific period of time. We have corrected the mistake and used “prevalence” in this article.
INTRODUCTION:
Q10:The introduction is directing the reader towards another objective than the one described, which would be to verify if patients with CL/P would have more depression than patients without CL/P. However, this objective is not described in the text.
A10:Thanks for your question. We have compared the difference of the results and added the results in table 2.
Q11: Are there studies that have analyzed which factors are associated with depression in patients with CL/P? Talk more about what already exists in the literature.
A11:Thank you for your comment, we have added related studies in the introduction.
Q12:Some results are not responding to the objective, so I am in doubt about whether some details of the method (not described) are relevant. I think it is important for the authors to define the role of control in this study and align with the objectives. Therefore, my comments will consider that this study intends to compare the prevalence of depression between patients with and without CL/P.
A12:Thanks for your question. We have compared the difference of the results and added the results in table 2.
Q13:Use the STROBE checklist to report your method, I miss some important details
A13:Thank you for your advice. We have used the STROBE checklist to report our methods.
Q14: Describe better about the objectives of the final study, considering that this publication is a preliminary one.
A14:Thank you for your advice. We have polished the objectives and added some clinical implications in the last paragraph of the introduction. The objectives of the final study have been described in the end of discussion.
Q15: Study design was not reported at any time.
A15:Thank you for your suggestions. We have added the study design.
Q16:Was the place of recruitment of the control group the same as the cases?
A16:Thank you for your question. Yes, the place of recruitment of the control group was the same as the study group. They were all the patients visiting our hospital in a certain period of time. We have pointed that out in the new version.
Q17:In this study, is there a need to pair the groups (despite the control group being smaller)? Explain more about it in the text.
A17: Thank you for your advice. We have explained more in the introduction.
Q18:What would be the diagnosis described in line 64.
A18: Thank you for your question. It was Individuals with no significant facial defects or other major diseases that may seriously affect physical or mental health.
Q19:Describe the response rate.
A19:Thank you for your advice. We have added the response rate in the Materials and Methods.
Q20: Make clear the groups used for analysis (It was not just with or without CL/P) and detail according to the statistical method. Some analyzes were only clear in the results section.
A20: Thank you for your advice. We have made the groups clear in the statistical analysis.
RESULTS
Q21: I suggest grouping the categories of the monthly family income variable, as some of them have a very low frequency.
A21:Thank you for your suggestion. The categories referred to the previous research (We have added the reference in the literature). However, it would not have a significant influence to our results as we used Pearson correlation analysis.
Q22:Was the analysis of patients with or without cleft performed using depression in a dichotomous way? With and without depression?
A22:Thank you for your question. Yes, all groups enrolled were analyzed in a dichotomous way.
Q23:In table 4, the CL group had a borderline P value for age. Discuss more about it.
A23:Thank you for your advise. We have added more discussion about the results.
Q24:I was confused by the scores in table 5. I went back to the method to understand but it does not describe.
A24: Thank you for pointing out this. We are sorry that some data were lost in the former version. The complete results were presented in the newly submitted version. TABLE 5 showed the results of the correlation between the monthly family income and the PHQ-9 scores in the study groups by Pearson correlation analysis.
DISCUSSION
Q25: Line 161-169. Discuss the findings. What do they mean?
A25:Thank you for your questions. We have adjusted the structure of the discussion. In this paragraph, we summarized the key points, and we discussed the findings respectively in the following paragraphs.
Q26: Line 170. As there are moments when the control group is compared with the CL/P and in others the analysis involves only the CL/P, make it clear that you are reporting the results of individuals with CL/P. (I reinforce the importance of aligning objectives with findings).
A26:Thank you for your advice. We have made the groups clear.
Q27: Did the authors not find other references to support the paragraph of my 184?
A27:Thank you for your advice. We have found others references and added them in the discussion.
Q28: Using the term items and not what they mean makes it very difficult to read the discussion. (line 193)
A28:Thank you for pointing out the question. We have added the meaning of the items in the discussion.
Q29: I am in doubt about the relevance of the result of the analyzes that were justified in the discussion simply as a lack of sample. Should they be there then? Or were the authors unable to discuss their findings? (Would grouping categories not help?)
A29:Thank you for your comments. We have added more explanation to it.
Reviewer 2 Report
Dear authors.
Thank you for allowing me to revise your work.
It is a sound study.
There are some aspects that I would like you to consider.
Introduction
- Paragraph 1 and 2 seems to jump from CLP to depression rather quickly.
- Adding a separate paragraph of why it is important to undertake this study is important. You have the components but, it needs to be explicitly stated.
Methodology
- What checklist did you follow?
- What theoretical framework did you approach
Results were sound but there are some results that is a bit heavy handed. For the interest of the readers it would be nice to highlight the most important aspect and refer the not so relevant results in the appendix. The readers will not have enough time to read over every aspects.
Discussion
- Please tighten the discussion. It has all the components but, it would flow better in the structure shown as below:
- Please structure your discussion into:
- Paragraph 1 - Summarise key points
- Paragraphs 2, 3 and 4 - Discuss, contextualize, theorise, put in the context of other work, why are your findings similar or different.
- Paragraph 5 - Implications for practice
- Paragraph 6 - Limitations and strengths
Author Response
Introduction
Q1: Paragraph 1 and 2 seems to jump from CLP to depression rather quickly.
A1:Thank you for your comment. We have added contents for better connections.
Q2: Adding a separate paragraph of why it is important to undertake this study is important. You have the components but, it needs to be explicitly stated.
A2:Thank you for your suggestion. We have added more clinical implications and emphasized the importance of the study in the introduction.
Methodology
Q3 What checklist did you follow?
A4:Thank you for pointing out the question. We followed the STROBE checklist for cross-sectional studies. And we have pointed that out in the methods.
Q4: What theoretical framework did you approach
A4: Thank you for your question. Some researchers have discovered the mental status of patients with CL/P. But the lack of a general view of the mental health of Chinese patients with CL/P urged us to do such research to discover the difference between the depression of the patients and normal people and the possible influencing factors. It would be helpful in the psychological interventions of the teamwork treatment of CL/P.
Q5:Results were sound but there are some results that is a bit heavy handed. For the interest of the readers it would be nice to highlight the most important aspect and refer the not so relevant results in the appendix. The readers will not have enough time to read over every aspects.
A5:Thank you for your advice. We have highlighted the most important aspects in the results and moved the others to the supplemental materials.
Discussion
Q6:Please tighten the discussion. It has all the components but, it would flow better in the structure shown as below:
- Please structure your discussion into:
- Paragraph 1 - Summarise key points
- Paragraphs 2, 3 and 4 - Discuss, contextualize, theorise, put in the context of other work, why are your findings similar or different.
- Paragraph 5 - Implications for practice
- Paragraph 6 - Limitations and strengths
A6:Thank you for your advise. We have adjusted the structure according to your suggestions, but there are 6 paragraphs discussing the results because we analyzed several possible risk factors and we wanted to explain them comprehensively.
Round 2
Reviewer 1 Report
Reference the STROBE in the text and place the completed checklist as a supplementary file
The comparison between depression in patients with and without CL/P is still unclear in the objectives
Lines 216 and 217 - I believe the item number is not missing, the description is sufficient
Author Response
Q1:Reference the STROBE in the text and place the completed checklist as a supplementary file
A1:Thank you for your advice. We have added completed checklist in the supplementary materials and referenced it in the text.
Q2:The comparison between depression in patients with and without CL/P is still unclear in the objectives
A2:Thank you for your suggestions. We have added the demonstration of the comparison in the objectives.
Q3:Lines 216 and 217 - I believe the item number is not missing, the description is sufficient
A3:Thank you for your comment. We have added the item numbers.